# The positive effect of plant diversity on soil carbon depends on climate

Marie Spohn [1] ✉, Sumanta Bagchi [2], Lori A. Biederman [3], Elizabeth T. Borer [4], Kari Anne Bråthen [5], Miguel N. Bugalho [6], Maria C. Caldeira [7], Jane A. Catford [8,9], Scott L. Collins [10], Nico Eisenhauer [11,12], Nicole Hagenah[13], Sylvia Haider [11,14,15], Yann Hautier [16], Johannes M. H. Knops[17], Sally E. Koerner [18], Lauri Laanisto [19], Ylva Lekberg [20], Jason P. Martina [21], Holly Martinson [22], Rebecca L. McCulley [23], Pablo L. Peri[24], Petr Macek [25], Sally A. Power [26], Anita C. Risch [27], Christiane Roscher [11,28], Eric W. Seabloom [4], Carly Stevens [29], G. F. (Ciska) Veen [30], Risto Virtanen [31] & Laura Yahdjian[32]

Little is currently known about how climate modulates the relationship between plant diversity and soil organic carbon and the mechanisms involved. Yet, this knowledge is of crucial importance in times of climate change and biodiversity loss. Here, we show that plant diversity is positively correlated with soil carbon content and soil carbon-to-nitrogen ratio across 84 grasslands on six continents that span wide climate gradients. The relationships between plant diversity and soil carbon as well as plant diversity and soil organic matter quality (carbon-to-nitrogen ratio) are particularly strong in warm and arid climates. While plant biomass is positively correlated with soil carbon, plant biomass is not significantly correlated with plant diversity. Our results indicate that plant diversity influences soil carbon storage not via the quantity of organic matter (plant biomass) inputs to soil, but through the quality of organic matter. The study implies that ecosystem management that restores plant diversity likely enhances soil carbon sequestration, particularly in warm and arid climates.

Plant diversity is positively related with soil organic carbon (SOC) storage in many ecosystems[1–5]. The most likely reason for this is that plant diversity positively affects plant productivity[6–10], and hence the amount of organic carbon input to soil[1,2,4]. Most evidence about the effects of plant diversity on plant productivity and SOC storage has been obtained based on small-scale experiments manipulating plant species richness (e.g.[1,2,9]). These biodiversity experiments make it possible to isolate the effect of plant diversity on ecosystem properties over short periods (i.e., maximum of a few decades). Yet, biodiversity experiments, particularly those of short duration, might underestimate the effects of plant diversity on ecosystem processes[11], because the impacts of diversity on productivity tend to escalate over time[12]. Therefore, real-world gradients of biodiversity that have developed over millennia have advantages over biodiversity experiments for exploring the relationship between biodiversity and ecosystem functioning[11,13]. This is particularly the case in studies exploring variables that change slowly in response to shifts in vegetation, such as SOC content[14–16]. Furthermore, in biodiversity experiments, abiotic factors are usually chosen to vary as little as possible in order to isolate the effect of plant diversity on ecosystem properties, which is opposite of real-world conditions where variation in abiotic conditions fosters plant diversity. In natural systems, variation in biodiversity is non-randomly distributed across space and time, whereas species combinations are randomly assembled in many biodiversity experiments[17].

Most biodiversity experiments have been conducted at a single location. Therefore, little is known about how climate influences the effect of plant diversity on ecosystem functioning, and specifically on SOC storage. SOC storage is strongly affected by climate since both plant productivity, i.e., production of organic matter via photosynthesis, and organic matter decomposition are temperature- and soil moisture-dependent[18,19]. It could be that the relationship between plant diversity and SOC storage is also climate-dependent. While little is known about this to date, the very few studies on effects of climate on relationships between plant diversity and ecosystem processes, including SOC storage, indicate a tendency for stronger relationships under drier climates[20,21].

The purpose of this study was to understand how plant diversity and soil organic matter are related in grasslands spanning a wide range of climate conditions. We hypothesized that plant diversity positively influences SOC content through a positive effect on plant biomass and, thereby, plant organic matter inputs to soil (Fig. 1a). Furthermore, we expected that the relationship between plant diversity on SOC is stronger in arid than in more humid grasslands. To test this hypothesis, we studied 84 natural and semi-natural grassland sites on six continents that represent 19 grassland types (Fig. S1 and Table S1). At each

site, an average of 30 plots were examined. The sites are a part of the Nutrient Network Global Research Cooperative (https://nutnet.org), but they were not experimentally manipulated in any way at the time of data collection.

## Results and discussion

### Plant diversity and soil organic matter

Contrary to our hypothesis, we found that the Shannon index of plant diversity was not significantly correlated with plant biomass across the 84 grasslands ($P = 0.119$). This might be because of two antagonistic processes whose effects on plant biomass cancel each other out. While plant diversity can increase plant biomass due to complementary use of resources by different plant species, elevated biomass can cause species loss due to shading of smaller species by larger species[10]. Our result is in accordance with a study that found no significant relationship between productivity and plant species richness across 48 grasslands on five continents, representing a subset of the sites examined here[22]. Our finding that plant biomass was not significantly correlated with the Shannon index suggests that plant diversity is not related with SOC through the rate of aboveground biomass input to soils (Fig. 1a).

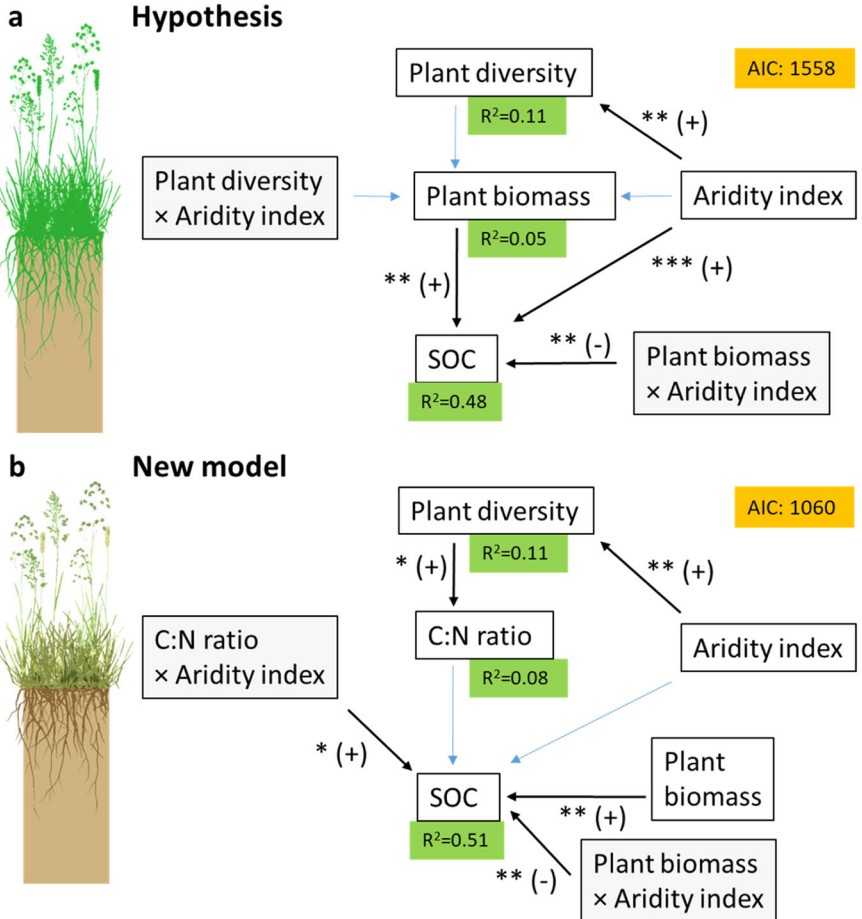

**Fig. 1 | Two structural equation models depicting the original hypothesis and the new, optimized model.** The initial hypothesis (**a**) states that plant diversity affects SOC through the quantity of organic matter (plant biomass) inputs to soil, whereas the new model (**b**) states that plant diversity affects SOC though the quality of organic matter (C:N ratio). Quality of plant organic matter is depicted in the drawing of the grassland in panel **b** by different colors. Gray boxes show interactions. Black arrows indicate significant regressions. Asterisks indicate the level of significance of the regressions (*$P < 0.05$, **$P < 0.01$, ***$P < 0.001$), and (+) and (-) indicate whether the slope of the linear regression model is positive or negative.

Blue arrows indicate non-significant regressions. The green boxes display the coefficient of determination ($R^2$) for the endogenous variables. The orange box displays the Akaike Information Criterion (AIC). The models were fitted to the site-level data. The new, optimized model (panel b) was obtained by increasing the model fit of the initial version of the new model (Fig. S6) by removing non-significant regressions. Plant diversity refers to the Shannon index. SOC stands for soil organic carbon. Plant drawings courtesy of Per-Marten Schleuss, used with permission.

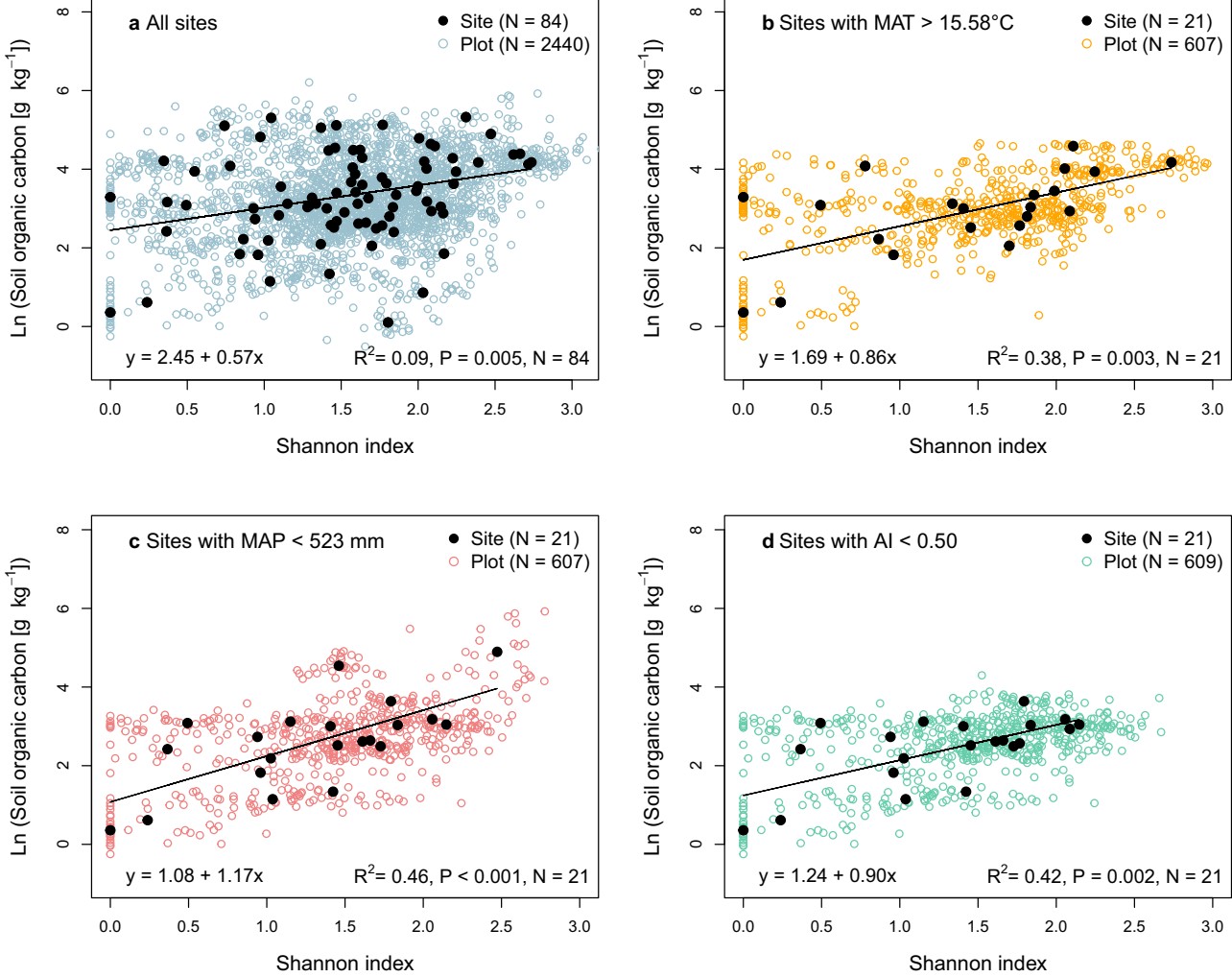

**Fig. 2 | Relationship between Shannon diversity index and soil organic carbon content.** The relationship is shown across all 84 grassland sites (**a**) as well as across sites with mean annual temperature (MAT) > 15.58 °C (**b**), sites with mean annual precipitation (MAP) < 523 mm (**c**), and arid and semi-arid sites, i.e., sites with an aridity index (AI) < 0.50 (**d**). The linear models were plotted to the site-level data (and not to the plot data, which is shown to give insight into the within-site variability). The subsets of sites shown in panels **b**, **c**, and **d** are the quartiles of sites for which significant correlations were found between Shannon index and soil organic carbon content (see Table 2). For further information on the relationship between Shannon index and soil organic carbon content depending on climate see Figure S2a, c, and e.

In contrast to plant biomass, SOC content was positively correlated with the Shannon index across all 84 sites ($P = 0.005$, $R^2 = 0.09$; Fig. 2a). Soil organic carbon content increased by a factor of 2.6 as the Shannon index increased from 0.0 (i.e., one plant species) to 2.5 (Fig. 2a). In three biodiversity experiments, the maximum increase in SOC caused by the largest increase in plant diversity was by a factor of 1.2[1,2] and 1.7[3]. Thus, the increase in SOC with increasing plant diversity observed here is comparatively large. It seems likely that biodiversity experiments underestimate the effect of plant diversity on SOC since the SOC content changes only slowly in response to shifts in vegetation, and typically requires several decades to reach a new steady state[14–16].

The soil carbon-to-nitrogen (C:N) ratio was positively correlated with the Shannon index across all 84 sites ($P = 0.006$, $R^2 = 0.09$; Fig. 3a). Plant diversity might affect the C:N ratio of plant biomass via shifts in the stem:leaf ratio potentially due to competition for light[23,24]. Plant individuals of the same species have been shown to increase in height with increasing diversity[25]. Taller plants have a higher stem:leaf ratio and thus produce more structural biomass[23,24] which has a high C:N ratio[26]. Our finding that C:N ratio increased with increasing Shannon index is in accordance with reported higher C:N ratios of aboveground biomass with greater plant species richness in a grassland biodiversity experiment in the central plains of North America[27]. Our findings also agree with the observation that increased species richness led to an increase in soil C:N ratio in a grassland experiment in the Netherlands[1].

Plant diversity might influence SOC content via the quality (C:N ratio) of organic matter. Organic matter with a high C:N ratio decomposes slowly due to its low nutritional value for microorganisms[28–30]. Thus, the increase in C:N ratio of plant biomass with higher plant diversity likely decreases the decomposition rate, and hence positively affects SOC content. This is supported by a litter decomposition study, conducted on a subset of the grassland sites examined here, which found that nitrogen addition to plant litter, which lowers the C:N ratio, increased early-stage litter decomposition[31]. Furthermore, compounds that give rigidity to the stem, such as lignin, typically decompose only slowly due to their complex structure[32], which could also contribute to the elevated SOC levels at sites with high plant diversity. Our results are in accordance with the observation that C:N ratio and SOC content are positively related across soils globally[33].

Plant diversity can affect not only the C:N ratio and the composition but also the diversity of organic compounds in plant litter and soils[34], which might also influence the decomposition rate, and hence

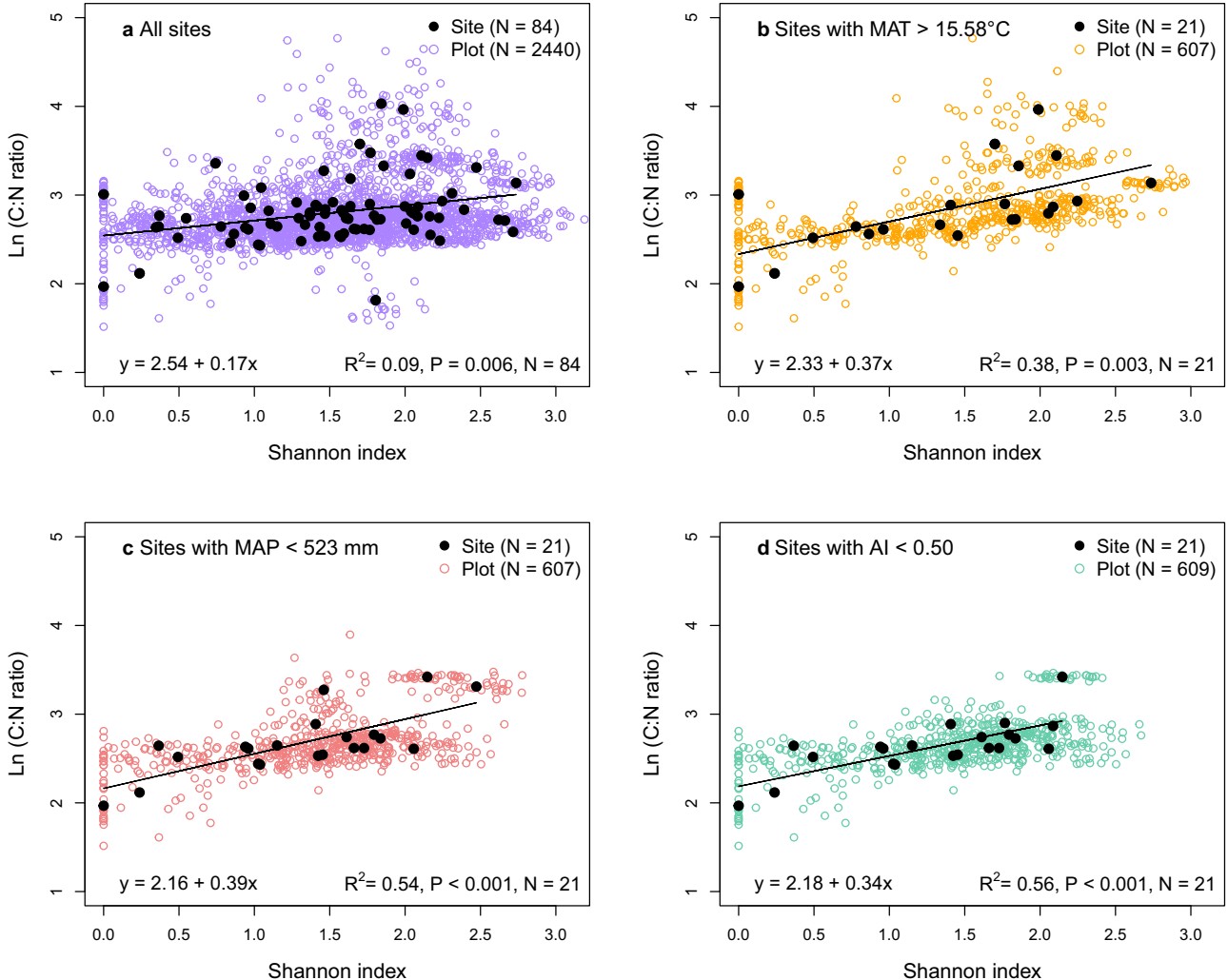

**Fig. 3 | Relationship between Shannon diversity index and soil C:N ratio.** The relationship is shown across all 84 grassland sites (**a**) as well as across sites with mean annual temperature (MAT) > 15.58 °C (**b**), sites with mean annual precipitation (MAP) < 523 mm (**c**), and arid and semi-arid sites, i.e., sites with an aridity index (AI) < 0.50 (**d**). Note that by definition the aridity index increases with decreasing aridity. The linear models were plotted to the site-level data (and not to the plot data, which is shown to give insight into the within-site variability). The subsets of sites shown in panels **b**, **c**, and **d** are the quartiles of sites for which significant correlations were found between Shannon index and soil C:N ratio (see Table 3). For further information on the relationship between Shannon index and soil C:N ratio depending on climate see Figure S2b, d, and f.

the SOC content[35]. The reason for this is that a greater diversity of molecules increases the cost of decomposition for soil microorganisms, as it requires a large suite of different enzymes[35,36]. Thus, a very diverse pool of organic compounds, derived from a diverse plant community, might decompose more slowly than a less diverse pool of organic matter from a less diverse community[34–36]. However, the positive relationship between SOC and plant diversity could potentially also result from other mechanisms. For instance, high plant diversity can lead to high soil microbial biomass[2,37] and soil aggregation[38], both of which can promote SOC sequestration[2,37,38].

### The effect of climate on the interaction of plant diversity and SOC

The positive correlation between Shannon index and SOC depended on climatic conditions (Table 1). Therefore, we divided the dataset into four equally-sized groups (quartiles) according to MAT, MAP and aridity index, following similar approaches in ecology and soil science (see Material and Methods). We only found a significant correlation between Shannon index and SOC for the quartiles (of sites) with the highest MAT (MAT > 15.58 °C; $P = 0.003$, $R^2 = 0.38$; Fig. 2b) or the lowest MAP (MAP < 523 mm; $P < 0.001$, $R^2 = 0.46$; Fig. 2c), and for the

quartile that only comprises arid and semi-arid sites (aridity index < 0.50; $P = 0.002$, $R^2 = 0.42$; Fig. 2d), but not across the sites that have a cooler and moister climate (Table 2). The strength of the relationship between Shannon index and SOC decreased with decreasing MAT (Fig. S2a) as well as with increasing MAP (Fig. S2c) and increasing aridity index (Fig. S2e; note that the aridity index increases by definition with decreasing aridity). At sites with high MAT, low MAP or low aridity index, a low Shannon index was on average associated with a lower SOC content than across all sites (compare intercepts in Fig. 2), and SOC content increased more with increasing Shannon index at these subsets of sites than across all sites (compare slopes in Fig. 2).

Likewise, the positive correlations between Shannon index and soil C:N ratio depended on climatic conditions (Table 1), and it was also stronger at warm and arid sites than across all sites (Fig. 3). We only found a significant correlation of Shannon index and soil C:N ratio for the quartiles of sites with the highest MAT (MAT > 15.58 °C; $P = 0.003$, $R^2 = 0.38$; Fig. 3b) or the lowest MAP (MAP < 523 mm; $P < 0.001$, $R^2 = 0.54$; Fig. 3c), but not across sites with lower MAT or higher MAP (Table 3). Further, there was a significant correlation of Shannon index and soil C:N ratio for the two quartiles of sites with lowest aridity index ($P < 0.001$, $R^2 = 0.56$ and $P = 0.041$, $R^2 = 0.20$, respectively, Fig. 3 and

**Table 1 | Results of multiple regression analyses**

| Regression | P value | Adjusted R² | P value Shannon | P value climate variable | P value interaction: Shannon × climate variable | N |
|---|---|---|---|---|---|---|
| Ln(SOC) ~ Shannon × MAT | <0.001 | 0.20 | 0.921 | 0.015 | 0.264 | 84 |
| Ln(SOC) ~ Shannon × MAP | <0.001 | 0.28 | <0.001 | <0.001 | **0.005** | 84 |
| Ln(SOC) ~ Shannon × AI | <0.001 | 0.42 | 0.003 | <0.001 | **0.011** | 84 |
| Ln(C:N ratio) ~ Shannon × MAT | 0.007 | 0.11 | 0.521 | 0.029 | **0.031** | 84 |
| Ln(C:N ratio) ~ Shannon × MAP | 0.002 | 0.12 | 0.002 | 0.019 | **0.034** | 84 |
| Ln(C:N ratio) ~ Shannon × AI | <0.001 | 0.16 | <0.001 | 0.002 | **0.006** | 84 |
| Ln(Biomass) ~ Shannon × MAT | 0.253 | - | 0.879 | 0.654 | 0.364 | 74 |
| Ln(Biomass) ~ Shannon × MAP | 0.011 | 0.11 | 0.107 | 0.037 | 0.218 | 74 |
| Ln(Biomass) ~ Shannon × AI | 0.038 | 0.07 | 0.117 | 0.042 | 0.178 | 74 |

Significant interactions ($P < 0.05$) between Shannon diversity index and climate variables are marked in bold font. The adjusted R² is given for all significant ($P < 0.05$) regressions. *SOC* Soil organic carbon, *Shannon* Shannon diversity index, *MAT* mean annual temperature, *MAP* mean annual precipitation, and *AI* aridity index. *N* refers to the number of observations (i.e. number of grassland sites).

**Table 2 | Results of regression analyses of the Shannon diversity index (Shannon) and soil organic carbon (SOC) content across the grassland sites in each of the four quartiles of mean annual temperature (MAT), mean annual precipitation (MAP), and aridity index (AI)**

| | 1. Quartile | 2. Quartile | 3. Quartile | 4. Quartile |
|---|---|---|---|---|
| MAT ranges (°C) of quartiles | −7.57–7.30 | 7.31–10.54 | 10.545–15.579 | 15.58–24.45 |
| Regression: Shannon and SOC for quartiles of MAT | P = 0.217 | P = 0.752 | P = 0.582 | **P = 0.003** **R² = 0.38** |
| MAP ranges (mm) of quartiles | 192–522.9 | 523–802 | 803–1054 | 1055–2566 |
| Regression: Shannon and SOC for quartiles of MAP | **P < 0.001** **R² = 0.46** | P = 0.765 | P = 0.409 | P = 0.897 |
| AI ranges of quartiles | 0.110–0.500 | 0.501–0.780 | 0.781–1.080 | 1.081–2.710 |
| Regression: Shannon and SOC for quartiles of AI | **P = 0.002** **R² = 0.42** | P = 0.392 | P = 0.688 | P = 0.084 |

The number of grassland sites in each quartile is 21 ($N = 21$). The R² is given for all significant ($P < 0.05$) regressions. Significant relationships ($P < 0.05$) are marked in bold font. Note that by definition the aridity index increases with decreasing aridity.

**Table 3 | Results of regression analyses of Shannon diversity index (Shannon) and soil organic carbon-to-nitrogen (C:N) ratio across the grassland sites in each of the four quartiles of mean annual temperature (MAT), mean annual precipitation (MAP), and aridity index (AI)**

| | 1. Quartile | 2. Quartile | 3. Quartile | 4. Quartile |
|---|---|---|---|---|
| MAT ranges (°C) of quartiles | −7.57–7.30 | 7.31–10.54 | 10.545–15.579 | 15.58–24.45 |
| Regression: Shannon and C:N ratio for quartiles of MAT | P = 0.372 | P = 0.578 | P = 0.568 | **P = 0.003** **R² = 0.38** |
| MAP ranges (mm) of quartiles | 192–522.9 | 523–802 | 803–1054 | 1055–2566 |
| Regression: Shannon and C:N ratio for quartiles of MAP | **P < 0.001** **R² = 0.54** | P = 0.171 | P = 0.075 | P = 0.872 |
| AI ranges of quartiles | 0.110–0.500 | 0.501–0.780 | 0.781–1.080 | 1.081–2.710 |
| Regression: Shannon and C:N ratio for quartiles of AI | **P < 0.001** **R² = 0.56** | **P = 0.041** **R² = 0.20** | P = 0.834 | P = 0.278 |

The number of grassland sites in each quartile is 21 ($N = 21$). The R² is given for all significant ($P < 0.05$) regressions. Significant relationships ($P < 0.05$) are marked in bold font. Note that by definition the aridity index increases with decreasing aridity.

Table 3). The strength of the relationship between Shannon index and C:N ratio decreased with decreasing MAT (Fig. S2b) as well as with increasing MAP (Fig. S2d) and increasing aridity index (Fig. S2f).

Our finding that there is only a significant relationship between Shannon index and SOC across the warm and arid sites, but not across the cooler and moister sites (Fig. 2 and Table 2) indicates that plant diversity affects SOC only under specific climate conditions. The most likely reason for this is that there is only a significant relationship between Shannon index and soil C:N ratio across the warm and arid sites, but not across the cooler and moister sites (Fig. 3 and Table 3). The higher soil C:N ratio likely leads to lower decomposition of organic matter, and thus to larger SOC contents[28–31] at the warm and arid sites, as discussed above (see previous section). The increase in C:N ratio

with increasing plant species diversity is likely the main reason for the positive relationship between plant diversity and SOC at the warm and arid sites.

There are also several other potential explanations why the relationship between plant diversity and SOC content is stronger at the warm and arid sites than across all sites. Plants at warm and arid sites produce more complex and difficult to decompose compounds, such as waxes, as protection against desiccation and solar radiation[39]. The decomposition of these complex compounds, which contain no nitrogen, is energetically unrewarding for microorganisms[35,36], and the diversity of complex compounds, together with the C:N ratio might increase with increasing plant diversity. Thus, complex compounds whose diversity increases with plant diversity could be a main reason

why the relationship between plant diversity and soil organic matter is particularly strong at warm and arid sites. Previous research on the relationship between chemical diversity of plant litter and decomposition reported contrasting findings[40], which might be because of the climate-dependence of the effect of litter diversity on decomposition. This is corroborated by a recent litter decomposition experiment conducted along a precipitation gradient in Chile, which found that mixing litters of several plant species led to a negative effect on decomposition only at the arid end of the precipitation gradient[41]. This result supports the interpretation that in warm and arid regions, chemical diversity of litter which increases with plant species diversity causes the strong relationship between plant diversity and SOC found here. Furthermore, it might even be that reduced decomposition, resulting from a high diversity of complex compounds, causes the elevated soil C:N ratios at sites with high plant diversity. However, it could also be that the relationship of plant diversity and SOC is comparatively large in warm and arid grassland soils since the effect of plant diversity on microbial biomass and soil aggregation is larger under specific climatic conditions, for unknown reasons. Our findings that the relationship between plant diversity and SOC content was stronger at arid and warm sites is in accordance with studies on Chinese grasslands[21] and European forests[20].

### Plant biomass, soil carbon, and climate

Plant biomass was positively correlated with SOC content across all 84 sites ($P = 0.008$, $R^2 = 0.09$; Fig. S3a) and the relationship between plant biomass and SOC depended on climatic conditions (Table S2). The relationship between SOC content and plant biomass was stronger across sites with high MAT (MAT > 15.58 °C; $P = 0.018$, $R^2 = 0.30$; Fig. S3b) than across all sites. SOC and plant biomass were not significantly correlated across sites with MAP < 523 mm or aridity index < 0.50 ($P = 0.702$ and 0.574, respectively), in contrast to the relationship observed between Shannon index and SOC (Fig. 2). The likely reason for this is that both aridity and low temperatures lead to low decomposition rates[19,42]. Thus, at arid and cooler sites, SOC content is probably more strongly shaped by the decomposition rate than by the rate of organic carbon input to soil[43], and consequently plant biomass and SOC are not correlated at these sites.

Across all 84 grassland sites, SOC was negatively correlated with MAT ($P < 0.001$, $R^2 = 0.19$; Fig. 4a) and positively with MAP ($P < 0.001$, $R^2 = 0.17$; Fig. 4b) and aridity index ($P < 0.001$, $R^2 = 0.38$; Fig. 4c; note that the aridity index increases by definition with decreasing aridity). The negative correlation of SOC and MAT suggests that decomposition increases more strongly than plant productivity with rising MAT across the 84 grasslands. Our finding is in accordance with previous studies reporting a negative relationship between SOC and MAT for soils in the US Great Plains[44] as well as soils in Australia[45]. Furthermore, our results are in agreement with a positive relationship between SOC and MAP in grassland soils in the US[46]. Yet, our analysis goes beyond these studies conducted at regional to national scales since it establishes relationships between MAT, MAP, the aridity index, and SOC across grasslands on six continents, and further elucidates the role of plant diversity for SOC storage.

SOC was more strongly correlated with aridity index than with MAT and MAP (Fig. 4). The reason for this is likely that aridity index is directly related to the mean soil moisture content[47] which, in turn, strongly influences plant productivity as well as organic matter decomposition[18,19]. The positive relationship between SOC and aridity index indicates that plant productivity increases more strongly than decomposition with declining aridity in grasslands. In contrast, the soil clay content was not significantly correlated with SOC content ($P = 0.864$), emphasizing the importance of climate over soil texture as a major control of soil organic content at the global scale, in accordance with the current view on the main determinants of SOC sequestration at different spatial scales[48]. Furthermore, soil C:N ratio

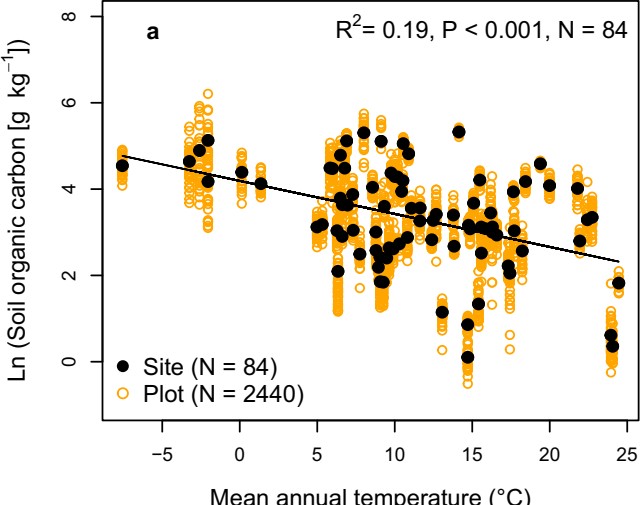

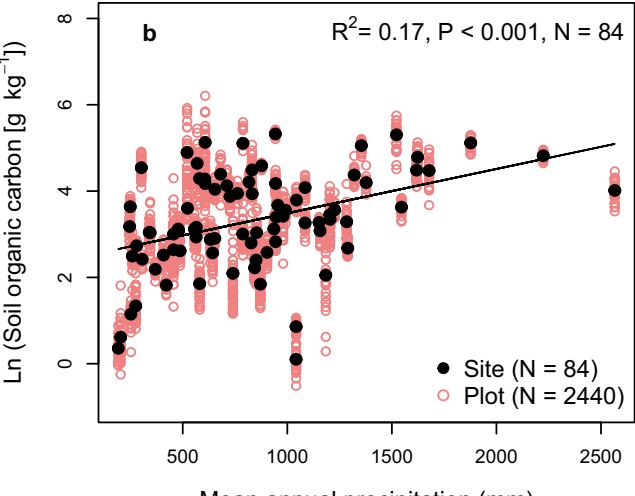

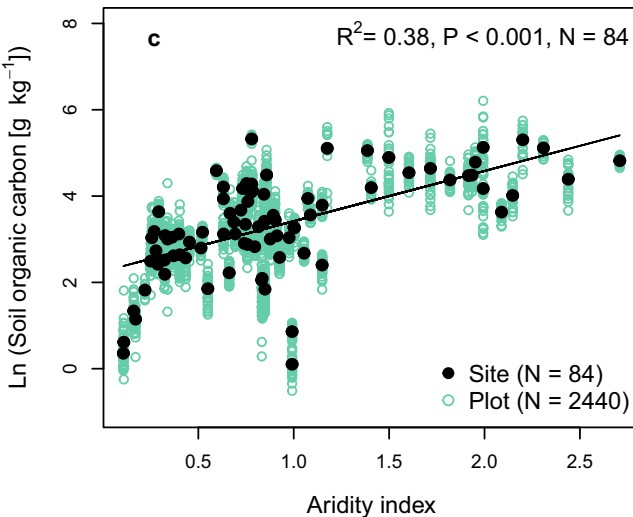

**Fig. 4 | Soil organic carbon content as a function of climate.** Soil organic carbon content as a function of mean annual temperature (**a**), mean annual precipitation (**b**), and aridity index (**c**) across 84 grasslands. Note that by definition the aridity index increases with decreasing aridity. The linear models were plotted to the site-level data (and not to the plot data, which is shown to give insight into the within-site variability).

was not significantly correlated with MAT and MAP, and only weakly with the aridity index ($P = 0.046$, $R^2 = 0.05$; Fig. S4), indicating that there is no consistent effect of MAT and MAP on C:N ratio of grassland soils globally. MAT and MAP were not significantly correlated ($P = 0.365$), but MAP was positively correlated ($P < 0.001$, $R^2 = 0.52$) and MAT negatively correlated ($P < 0.001$, $R^2 = 0.22$) with the aridity index (which by definition decreases with increasing aridity).

## Modeling interactions between plants and soil organic matter

A model predicting that plant diversity is related to SOC through its relationship with soil C:N ratio in interaction with aridity (Fig. 1b) fit our data better than a model predicting that plant diversity is related to SOC through its relationship with plant biomass (as stated in our original hypothesis; Fig. 1a). The new model (Fig. 1b) supports the concept that plant diversity influences SOC sequestration through organic matter quality and not via quantity of plant biomass inputs to soil. While plant biomass was significantly correlated with SOC, it was not significantly correlated with plant diversity ($P = 0.119$). The model (Fig. 1b) predicts that both the C:N ratio and plant biomass affect SOC content in interaction with aridity, confirming that the relationship between plant diversity and SOC is modulated by climate.

There are several feedbacks (bidirectional interactions) between soil organic matter and plants. In this study, we concentrated on the effect of plants on soil organic matter, which does not rule out a potential influence of soil organic matter on plant biomass or diversity. Soil organic matter contains nutrients and can positively affect soil water holding capacity, which, in turn, can influence plant biomass and diversity. To disentangle the causalities in the feedback between soil and plants and evaluate the dominant direction of the causalities, each variable in this interaction would have to have a sufficient degree of independence in its predictors. We found that the positive correlation of soil nitrogen and Shannon index (Fig. S5) was less strong than the one of SOC and Shannon index (Fig. 2), which is in accordance with earlier work[1]. In addition, we observed that soil phosphorus, which can also be bound in organic matter, was not significantly correlated with the Shannon index ($P = 0.914$). Together, these findings suggest that the effect of soil nutrients on plant diversity is smaller than the effect of plant diversity on SOC that is likely caused by a change in the quality of the organic matter, leading to reduced organic matter decomposition (see above). Still, it is likely that soil organic matter and the nutrients therein (particularly nitrogen) also affect plant diversity. A bidirectional, positive interaction between soil organic matter and plant diversity, developed over several decades to centuries[14–16], likely explains the strong relationship between plant diversity and SOC found in this study compared to biodiversity experiments[1–3].

## The effect of plant diversity on soil carbon

In conclusion, we found support for the first part of our hypothesis that plant diversity is positively related with SOC content, and we observed that this relationship was strongest in arid and warm grasslands. However, we did not find support for the second part of the hypothesis that plant diversity affects SOC content through a positive effect on plant biomass (Fig. 1a). Instead, we present evidence indicating that plant diversity impacts SOC through its effect on C:N ratio, in a manner that depends on climate (Fig. 1b). Thus, our results suggest that SOC content is affected by plant diversity through organic matter quality, while plant aboveground biomass, i.e., quantity, is also related to SOC but not to plant diversity. In more general terms, our study demonstrates that the relationship between biodiversity and ecosystem processes is climate-dependent, which is crucial for understanding ecosystem functioning and emphasizes the importance of moving beyond local experiments in ecology. Our study has important implications since grassland ecosystems store approximately one third of the global terrestrial carbon stocks[49]. Thus, potential future losses of plant diversity in grasslands could jeopardize SOC storage, particularly in warm and arid climates.

## Methods

### Study sites

The 84 grassland sites explored in this study are natural and semi-natural grasslands located on six continents, covering a wide range of climatic conditions (Table S1, Fig. S1). Across the grassland sites, MAT ranges from −7.57 °C to 24.45 °C, MAP ranges from 192 mm to 2566 mm, and the aridity index ranges from 0.107 to 2.709. The 84 sites represent 19 grassland types (Table S1, Fig. S1). All 84 sites are part of the Nutrient Network Global Research Cooperative[50] (NutNet, https://nutnet.org). Nine of the 84 sites, located on different continents and in different climate zones, are classified as the grassland type old field (Table S1), and they likely received some fertilizer in the past. Yet, the large majority of the sites has never received any fertilizer. Furthermore, nutrient contents differ widely among the 84 grasslands, independently of former land-use. For this study, we choose data that were collected in the year before any experimental treatment started, which means that the sites were not experimentally manipulated at the time of data collection.

### Sampling, measurements, and climate data

At each site, plots measuring 5 × 5 m were established. At most sites, 30 of these plots were established, with the number of plots ranging from 10 to 60 (Table S1). All sites followed the same experimental sampling protocol, and the data were collected between 2007 and 2020.

Plant species diversity (called plant diversity hereafter) was determined in a randomly designated 1 × 1 m subplot within each 5 × 5 m plot at peak biomass. In the 1 × 1 m subplot, cover was estimated visually to the nearest 1% for every species overhanging the subplot. Data on plant diversity were collected at all 84 sites.

Live vascular plant aboveground biomass (called plant biomass hereafter) was estimated destructively by clipping at ground level all aboveground biomass of plants rooted within two 1 × 0.1 m strips (for a total of 0.2 m²) adjacent to the 1 × 1 m subplot where plant species diversity was determined. All biomass was dried at 60 °C to constant mass before weighing to the nearest 0.01 g. Data on plant biomass were collected at 74 of the 84 sites.

Soil samples were collected in the 5 × 5 m plots by taking three soil cores (2.5 cm diameter) at a depth of 0–10 cm. The three cores were pooled to make one sample per plot. Root fragments were removed, the soils were air-dried, and sieved (<2.0 mm) prior to any analysis. The samples were analyzed for total organic carbon (called soil organic carbon or soil carbon hereafter) and total nitrogen using an elemental analyzer (Costech ECS 4010 CHNSO Analyzer). Plant-available phosphorus and was extracted from soil according to the Mehlich-3 protocol[51] and quantified using Inductively Coupled Plasma Mass Spectrometry. Soil texture was measured using the Bouyoucos method. All soil samples were analyzed in the same laboratory (A&L Analytical Laboratory). Data on soil organic carbon (SOC) and soil nitrogen and soil phosphorus were collected at all 84 sites, and data on soil texture were collected at 62 sites.

We obtained data on mean annual temperature (MAT) and mean annual precipitation (MAP) from Worldclim for all 84 sites[52]. In addition, we obtained data on potential evapotranspiration (PET) from the Consultative Group for International Agricultural Research (CGIAR) for all 84 sites.

### Calculations and data analysis

The aridity index was calculated by dividing MAP by PET. By definition, the aridity index increases with decreasing aridity[53]. An aridity index of <0.05 indicates hyperarid climate, 0.05–0.20 arid climate, 0.20–0.50 semi-arid-climate, and 0.50–0.65 dry-subhumid climate. The Shannon-Wiener diversity index (called Shannon index and Shannon diversity index hereafter) was calculated from the plant diversity data collected at the plot scale using the R package VEGAN (version 2.6-4)[54]. The molar organic carbon-to-nitrogen (C:N) ratio was

calculated by dividing the moles of soil organic carbon by the moles of soil nitrogen.

We calculated arithmetic means of the Shannon index, SOC, soil nitrogen, soil C:N ratio as well as plant biomass across all plots at each site. Regression analyses were conducted based on the means calculated for each site (and not based on the plot-level data since the observations for different plots of one site are not independent of each other). SOC, soil nitrogen, and soil C:N ratio were not normally distributed and were therefore transformed by calculating their natural logarithm prior to regression analysis. We calculated linear regression models for SOC, C:N ratio, plant biomass, and the Shannon index as functions of MAT, MAP and aridity index across all sites. Further we calculated linear regression models for plant biomass, SOC, soil nitrogen, C:N ratio, and soil phosphorus as a function of the Shannon index as well as SOC as a function of plant biomass or soil clay content. We report the coefficient of determination ($R^2$) for all significant ($P < 0.05$) regressions.

We conducted multiple regression analysis to investigate whether the relationship between Shannon index and SOC depended on climatic conditions (mean annual temperature, mean annual precipitation, and aridity index). In addition, we conducted multiple regression analysis to explore whether the relationships between Shannon index and soil C:N ratio, Shannon index and plant biomass as well as plant biomass and SOC depended on climatic conditions.

We then conducted further regression analyses to investigate how the strength of the relationships between Shannon index and SOC, Shannon index and soil C:N ratio, as well as plant biomass and SOC varied with climatic conditions. For this purpose, we first divided the 84 sites into four quartiles (each containing 21 sites), according to their MAT, MAP or aridity index (separately for each of the three climate variables). Second, we calculated regressions across the sites of each quartile. This is the most objective way of creating subsets of sites, and we adapted this approach of dividing the dataset into subsets of sites from previous studies. For instance, Wang et al. (2019) divided their dataset on grassland ecosystems into three equally-sized subsets of sites according to plant productivity (low, medium, and high productive sites)[55]. Malik et al. (2018) divided their dataset on SOC dynamics into subsets of sites according to soil pH[56]. Further, Starke et al. (2020) divided their dataset on plant cover and soil erosion into subsets according to latitude[57].

Each subset (quartile) has a broad representation of both grassland type and region. The subset (quartile) of sites with highest MAT (MAT > 15.58 °C) includes sites from North America, Australia, Europe, South America, and Africa. The subset (quartile) of sites with lowest MAP (MAP < 523 mm) also includes sites from North America, Australia, Europe, South America, and Africa. The subset (quartile) of sites with lowest aridity index (i.e., arid and semi-arid climate; aridity index <0.50) includes sites from North America, Australia, Europe, and South America (see Table S1).

Since we found that Shannon index was only significantly correlated with SOC across the quartile of sites with the highest MAT, the lowest MAP or the lowest aridity index (Table 2), but not across the cooler and moister sites, we conducted further regression analyses (Fig. S2). The purpose of these additional analyses was to investigate how the relationships between Shannon index and SOC as well as Shannon index and C:N ratio change with climate. For this purpose, we created subsets of sites chosen according to their MAT, MAP or aridity index. The smallest subset consisted of the eight sites with either the highest MAT, lowest MAP or lowest aridity index. Subsequently, we increased the size of the subsets in a stepwise manner by decreasing MAT or increasing MAP or aridity index (adding one site per step). We calculated the regression models for Shannon index and SOC as well as Shannon index and C:N ratio across the 26 different subsets of sites (Fig. S2) to analyze how the relationships between Shannon index and SOC as well as Shannon index and C:N ratio change with climate.

## Piecewise structural equation modeling

We conducted piecewise structural equation modeling using the R package *piecewiseSEM* (version 2.3.0)[58]. We choose to conduct piecewise structural equation modeling because it allows us to include interactions between climate and ecosystem properties (in contrast to other path modeling approaches). We first depicted our hypothesis in a piecewise structural equation model. We fitted the site-level data to this model and evaluated the model fit using the Akaike Information Criterion (AIC). Based on the results of the regression analyses, which were not in accordance with the second part of our hypothesis, we formulated an initial version of a new model (Fig. S6). We fitted the data to this model and evaluated the fit of the model based on AIC. Subsequently, we improved the fit of the new model to the data by stepwise removal of non-significant regressions, and we evaluated the fit of each resulting new model based on AIC. We stopped this process when any further removal of a regression did not lead to a decrease in AIC. All data analyses were conducted using R (version 4.2.1)[59].

## Reporting summary

Further information on research design is available in the Nature Portfolio Reporting Summary linked to this article.

## Data availability

All data are available at this repository. https://doi.org/10.5281/zenodo.8308135.

## Code availability

All R code for reproducing the results is available at this repository. https://doi.org/10.5281/zenodo.8308135.

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

## Acknowledgements

This work was generated using data from the Nutrient Network (http://www.nutnet.org) experiment, funded at the site-scale by individual researchers. Coordination and data management have been supported by funding to E.T.B. and E.W.S. from the National Science Foundation Research Coordination Network (NSF-DEB-1042132) and Long Term Ecological Research (NSF-DEB-1234162 to Cedar Creek LTER) programs, and the Institute on the Environment (DG-0001-13). We also thank the Minnesota Supercomputer Institute for hosting project data and the Institute on the Environment for hosting Network meetings. Soil analyses were supported by funds from Oregon State University and University of Minnesota to E.T.B. and E.W.S. and by USDA-ARS grant 58-3098-7-007 to E.T.B. M.S. thanks Björn Lindahl for helpful comments on a previous version of the manuscript and Per-Marten Schleuss for the drawings in Fig. 1. M.C.C. and M.N.B. gratefully acknowledge the Portuguese Science Foundation (FCT) for funding the research units CEF (UIDB/00239/2022) and CEABN-InBIO (UID/BIA/50027/2020), and thank Rui Alves for logistic support and for granting access to Companhia das Lezirias study site. L.L. is funded by Estonian Academy of Sciences (research professorship for Arctic studies). S.H. gratefully acknowledges the support of iDiv funded by the German Research Foundation (DFG– FZT 118, 202548816). S.L.C. acknowledges the support of NSF-1856383. Y.L. is grateful to MPG Ranch for funding.

## Author contributions

M.S. developed and framed research questions, analyzed data, and wrote the paper. E.T.B. and E.W.S. are Nutrient Network Coordinators and site level coordinators, and contributed to paper writing. S.B., L.A.B., K.A.B., M.N.B., M.C.C., J.A.C., S.C., N.E., N.H., S.H., Y.H., J.M.H.K., S.E.K., L.L., Y.L., J.P.M., H.M., R.L.M, P.L.P., P.M., S.A.P., A.C.R., C.R., C.S., G.F.V., R.V., L.Y. are site level coordinators and contributed to paper writing (see also Table S3).

## Funding

## Competing interests

The authors declare no competing interests.

## Additional information

[1]Department of Soil and Environment, Swedish University of Agricultural Sciences (SLU), Lennart Hjelms väg 9, 75007 Uppsala, Sweden. [2]Indian Institute of Science, Bangalore 560012, India. [3]Department of Ecology, Evolution, and Organismal Biology, Iowa State University, 251 Bessey Hall, Ames, IA 50011, USA. [4]Department of Ecology, Evolution, and Behavior, University of Minnesota, St Paul, MN, USA. [5]Department of Arctic and Marine Biology, UiT – Arctic University of Norway, Tromsø, Norway. [6]Centre for Applied Ecology "Prof. Baeta Neves" (CEABN-InBIO), School of Agriculture, University of Lisbon, Lisbon, Portugal. [7]Forest Research Centre, Associate Laboratory TERRA, School of Agriculture, University of Lisbon, Lisbon, Portugal. [8]Department of Geography, King's College London, 30 Aldwych, London WC2B 4BG, UK. [9]School of Agriculture, Food and Ecosystem Sciences, University of Melbourne, Parkville, VIC 3010, Australia. [10]Department of Biology, University of New Mexico, Albuquerque, NM 87131, USA. [11]German Centre for Integrative Biodiversity Research (iDiv) Halle-Jena-Leipzig, Puschstraße 4, 04103 Leipzig, Germany. [12]Leipzig University, Institute of Biology, Puschstraße 4, 04103 Leipzig, Germany. [13]Mammal Research Institute, Department of Zoology & Entomology, University of Pretoria, Pretoria, South Africa. [14]Leuphana University of Lüneburg, Institute of Ecology, Universitätsallee 1, 21335 Lüneburg, Germany. [15]Martin Luther University Halle-Wittenberg, Institute of Biology and Geobotany and Botanical Garden, Am Kirchtor 1, 06108 Halle, Germany. [16]Ecology and Biodiversity Group, Department of Biology, Utrecht University, Padualaan 8, 3584 CH Utrecht, The Netherlands. [17]Health and Environmental Sciences, Xián Jiaotong-Liverpool University, Suzhou, China. [18]Department of Biology, University of North Carolina Greensboro, Greensboro, NC, USA. [19]Department of Biodiversity and Nature Tourism, Estonian University of Life Sciences, Kreutzwaldi St. 5, 51006 Tartu, Estonia. [20]MPG Ranch and University of Montana, Montana, USA. [21]Department of Biology, Texas State University, San Marcos, TX 78666, USA. [22]Department of Biology, McDaniel College, Westminster, MD 21157, USA. [23]Department of Plant & Soil Sciences, University of Kentucky, Lexington, KY 40546, USA. [24]National Institute of Agricultural Technology (INTA), Rio Gallegos, Santa Cruz, Argentina. [25]Institute of Hydrobiology, Biology Centre of the Czech Academy of Sciences, Na Sadkach 7, 370 05 Ceske Budejovice, Czech Republic. [26]Hawkesbury Institute for the Environment, Locked Bag 1797, Penrith, NSW 2751, Australia. [27]Swiss Federal Institute for Forest, Snow and Landscape Research WSL, Zuercherstrasse 111, 8903 Birmensdorf, Switzerland. [28]UFZ, Helmholtz Centre for Environmental Research, Department Physiological Diversity, Permoserstrasse 15, 04318 Leipzig, Germany. [29]Lancaster Environment Centre, Lancaster University, Lancaster LA1 4YQ, UK. [30]Department of Terrestrial Ecology, Netherlands Institute of Ecology, Droevendaalsesteeg 10, 6708 PB Wageningen, The Netherlands. [31]Ecology & Genetics, University of Oulu, PO Box 3000, 90014 Oulu, Finland. [32]Instituto de Investigaciones Fisiológicas y Ecológicas Vinculadas a la Agricultura (IFEVA), CONICET, Faculty of Agronomy, University of Buenos Aires, Buenos Aires, Argentina. ✉e-mail: marie.spohn@slu.se

