## [Peer Review File · Nature Communications]

The positive effect of plant diversity on soil carbon depends on
climateReviewer #1 (Remarks to the Author):

The authors tried to interpret a new finding that plant diversity affects SOC through climate-dominated C:N ratio based on globally distributed grassland sites. They also found that the relationship between diversity and SOC is better in dry and warm grasslands. Although the work is potentially interesting, the manuscript is not ready for publication since several open questions remain.

My major concern is in the statistical analysis, further leading to the unreliability of conclusions in the article. The study determined the subset of climatic conditions (MAT, MAP or aridity index) influencing the relationships of diversity with SOC and soil C:N ratio based on the highest R². This can lead to different subsets containing different grassland sites for different climatic conditions, which in turn affects the comparability of results because of the sample dependency. In addition, the relatively small number of grassland sites affects the reliability of the conclusions. Moreover, the grassland sites in the paper are mainly distributed in North America and Europe, with a serious lack of sampling of grassland types such as Asia and South America. This is reflected in the statistics that the sample sites in a given climate may all be taken from similar regions, preventing global conclusions. I would like to suggest the authors to use bootstrapping method or to analyze the same subset based on the same climatic zone.

Most explanations of the conclusions appear to be based on probable causes without direct data support. The interaction of vegetation, climate, and soil is context-dependent. Different vegetation types, the response of vegetation to climate extremes, and the legacy effect of past climate on current vegetation productivity may all influence the conclusions of the article.

Minor issues

- Line 69: grasslands or grassland sites
- Line 82: Mislabeled references
- Line 102: I recommend supplementing these grassland sites with grassland types
- Line 106: I recommend supplementing these grassland sites with grassland types
- Line 111: The relationships among plant species richness, productivity, SOC and C:N ratio across the 84 grasslands should be added at least in the supplement material.
- Line 112: Wang et al. (2019) indicated that effect of species richness on productivity shifts from strongly positive in low-productivity communities to strongly negative in high-productivity communities. We found that the article analyzed the relationship between diversity and SOC and C:N ratio in different climatic conditions, but did not analyze the relationship between diversity and biomass in different climatic conditions to better support the conclusions.
- Line 117-119: I am lost here. How the factor was calculated. Is it comparable at different diversity gradients
- Line 131: Examples of forest ecosystems are not relevant.
- Line 116-197: In Fig.2 and Fig.3, I found that the regression analysis was fitted based on site data, not plot data, so that the role of the plot data is only to show? The smaller number of sample plots (N<17) may not support the findings of the paper. At least the error bar of each site should be shown. In addition, only a subset of the results meeting the climatic threshold conditions were shown, and another part should be shown to prevent bringing down the reader's understanding of how climate affects the diversity-SOC relationship
- Line 188: In Figure 4, the model is fitted with site data and the presentation of plots makes no sense. And all the plots of the same site seem to share one set of climatic data, please give the resolution of the climatic data.
- Line 216: Missing segment comma
- Line 238: "Evidence indicating that plant diversity impacts SOC through its effect on C:N ratio ". But the structural equation model in Figure 1 demonstrates that the pathway relationship from soil C: N to SOC was not significant.
- Line 466: Materials and methods line feed

Reviewer #3 (Remarks to the Author):

Thank you for the opportunity to review your paper. I appreciate the time and work that goes into

preparing a manuscript.

This manuscript uses comprehensive data from the NutNet and focuses on the role of background climate conditions in modulating the relationship between plant diversity and soil carbon. The structural equation modelling approach was used to detect the interaction effects of climate conditions and plant diversity on soil C. As a main result, the authors showed that plant diversity is positively related to soil carbon content across 84 grassland sites and this positive relationship is more pronounced in warm and arid climates. Importantly, the authors found that the quality of organic matter, instead of the quantity of organic matter, is the main underlying mechanism of the positive relationship between plant diversity and soil carbon.

Interesting read! The design of the study is quite impressive, the results are novel, interesting, and important. However, there are some concerns. Below, I listed some major points that should be addressed.

General Comments:

1. The authors stated that real-world observation research has advantages over biodiversity experiments for exploring the BEF relationship due to long-term developments. However, I do not agree with the long-term vs short-term is the main difference between experiments and natural observations. The main differences are randomly assembled species richness levels and uniform environment in experiments. Please see van der Plas, F. Biodiversity and ecosystem functioning in naturally assembled communities. *Biol. Rev.* 94, 1220–1245 (2019).

Thus the positive relationships in experiments might be not real.

2. The authors state that biodiversity experiments might underestimate the effect of plant diversity on SOC since they found more strong effects. However, it's worth considering that stands with higher nutrient levels (as indicated by high SOC) could potentially support a greater number of species. Therefore, it may be the soil nutrient levels influencing diversity rather than plant diversity directly impacting soils. In a recent study, it was found that the effect of diversity on soil carbon accumulation is positive but relatively weak.

Chen, X. Tree diversity increases decadal forest soil carbon and nitrogen accrual. *Nature*, 2023.

3. I very much like the new mechanism (litter C:N) that arises in this manuscript. However, it is important to note that this does not necessarily imply that only the quality, and not the quantity, of litter inputs influences SOC. It is plausible that the increased plant diversity resulted in enhanced root productivity, ultimately leading to an increase in soil carbon. Please see: Chen, S. P. et al. Plant diversity enhances productivity and soil carbon storage. *Proc. Natl Acad. Sci. USA* 115, 4027–4032 (2018).

4. I am curious about the inclusion of both site and plot data in Figure 2. Is it appropriate to pool them together? Furthermore, during the model selection process, did the authors take into account the possibility of multicollinearity?

Specific Comments:

Line 91 please see

Chen, X., Hisano, M., Taylor, A. R. & Chen, H. Y. H. The effects of functional diversity and identity (acquisitive versus conservative strategies) on soil carbon stocks are dependent on environmental contexts. *For. Ecol. Manag.* 503, 119820 (2022).

Line 151-152 Please note high plant diversity could SOC via soil microorganisms due to increased litter inputs, which is based soil microbial carbon pump hypothesis. Only increased litter C:N without any enhanced litter inputs might decrease soil activity and SOC from soil microbial residues. Please see:

Lange, M. et al. Plant diversity increases soil microbial activity and soil carbon storage. *Nat. Commun.* 6, 6707 (2015).

Replies to the reviewers

We thank both reviewers for their comments which helped us to improve the manuscript. Please find below our answers to the comments, written in blue (line numbers refer to the manuscript without tracked changes). Quotations from the manuscript are displayed in cursive font.

Reviewer #1: The authors tried to interpret a new finding that plant diversity affects SOC through climate-dominated C:N ratio based on globally distributed grassland sites. They also found that the relationship between diversity and SOC is better in dry and warm grasslands. Although the work is potentially interesting, the manuscript is not ready for publication since several open questions remain.

My major concern is in the statistical analysis, further leading to the unreliability of conclusions in the article. The study determined the subset of climatic conditions (MAT, MAP or aridity index) influencing the relationships of diversity with SOC and soil C:N ratio based on the highest R². This can lead to different subsets containing different grassland sites for different climatic conditions, which in turn affects the comparability of results because of the sample dependency.

We thank the reviewer for this comment which stimulated several improvements of our manuscript. In this manuscript, we explore the relationship between plant diversity and soil carbon storage based on grasslands located on six continents that span wide climatic gradients. Across all sites, mean annual temperature (MAT) ranges from -7.57 °C to 24.45 °C and mean annual precipitation (MAP) ranges from 192 mm to 2566 mm. We analyzed how climate modulates the relationship between plant diversity and soil carbon storage, and we found that the relationship between plant diversity and soil organic carbon content is strongest at sites with high MAT, low MAP, and strong aridity.

The segmented regression approach allows us to identify the conditions under which the relationship between plant diversity and soil organic matter are strongest. In the previous version of the manuscript, we had selected subsets of sites according to the highest coefficient of determination (R²). Following consultation with a statistician, we changed this method and used a more objective approach by subsetting into quartiles (according to climate variables). This approach increased the number of sites in most subsets, by up to 38%. Accordingly, we changed **Figures 2 and 3**. This part of the data analysis is described in the Material and Methods section **lines 329-339**.

Furthermore, we created a new six-panel figure (**Fig. S2 in the Supplement**) showing the results of an analysis exploring how changes in the size of the subsets affects the results of the regression analyses. This new figure supports our main result that the relationships between Shannon index and SOC as well as Shannon index and C:N ratio strongly depend on climate, and are strongest in warm and arid climates. Thus, this new, additional figure confirms that our conclusions are solid and statistically valid. This part of the data analysis is described in the Material and Methods section **lines 346-357**.

The reviewer's main concern here seems to be that there is a codependence of climate zones and regions as well as grassland types. In fact, the subsets of sites with highest MAT, lowest MAP, and lowest aridity index that we consider in the regression analyses include locations spanning 4-5 continents. Thus, the dataset on grassland sites analyzed here is not spatially restricted, and the spatial distribution of sites does not bias our conclusions. We now added the following sentences in the Material and Methods section (**lines 340-345**).

“The subset (quartile) of sites with highest MAT (MAT > 15.58 °C) includes sites from North America, Australia, Europe, South America, and Africa. The subset (quartile) of sites with lowest MAP (MAP < 523 mm) also includes sites from North America, Australia, Europe, South America, and Africa. The subset (quartile) of sites with arid and semi-arid climate (aridity index < 0.50) includes sites from North America, Australia, Europe, and South America (see also Table S1).”

In addition, the relatively small number of grassland sites affects the reliability of the conclusions. Moreover, the grassland sites in the paper are mainly distributed in North America and Europe, with a serious lack of sampling of grassland types such as Asia and South America. This is reflected in the statistics that the sample sites in a given climate may all be taken from similar regions, preventing

global conclusions. I would like to suggest the authors to use bootstrapping method or to analyze the same subset based on the same climatic zone.

Our dataset covers 84 grassland sites on six continents, and it includes 19 types of grasslands (mesic grassland, mixedgrass prairie, alpine grassland, old field, grassland steppe, shortgrass prairie, tallgrass prairie, montane grassland, semiarid grassland, calcareous grassland, annual grassland, desert grassland, pasture, shrub steppe, coastal grassland, tundra grassland, salt marsh, and savanna). At each of the 84 sites, on average 30 plots were analyzed. We are not aware of a stronger dataset for these analyses on the relationship between plant diversity and soil organic matter; ours includes all types of grasslands and contains data on all continents except for Antarctica. It is true that the dataset contains more data on European and North American sites than on Asian and South American sites. Nonetheless, for South America, the dataset contains data on four different grassland types (grassland steppe, mesic grassland, alpine grassland, semiarid grassland), representing the main grassland types on the South American continent.

It is not clear to us how bootstrapping could add to the analysis. We consulted with a statistician and now revised our approach to create subsets of sites according to climate variables (see Material and Methods **lines 329-339**). This increases the size of sites in most subsets, by up to 38% (for a description of the subsets, please see the cursive text in our last answer). We improved **Figures 2 and 3** accordingly.

In addition, we analyzed how the coefficient of determination (R^2) of linear models of Shannon index and soil organic carbon (SOC) as well as Shannon index and carbon-to-nitrogen (C:N) ratio varies with climate. For this purpose, we created a new six-panel figure (**Fig. S2 in the Supplement**). The figure shows the R^2 of the linear models of Shannon index and SOC (a, c, e) and Shannon index and C:N ratio (b, d, f) for different subsets as a function of the mean of the climate variable of each subset (mean annual temperature, mean annual precipitation or aridity index). The new figure demonstrates that the relationships between Shannon index and SOC as well as Shannon index and C:N ratio are strongest in warm and arid climates. The figure confirms that our conclusions are solid and statistically valid. This part of the data analysis is described in the Material and Methods section **lines 346-357**.

Furthermore, we moved two tables that used to be in the Supplement to the main part of the manuscript (now called **Tables 2 and 3**) since they show results of statistical analyses which further support our main finding.

Most explanations of the conclusions appear to be based on probable causes without direct data support. The interaction of vegetation, climate, and soil is context-dependent. Different vegetation types, the response of vegetation to climate extremes, and the legacy effect of past climate on current vegetation productivity may all influence the conclusions of the article.

We are unclear what the reviewer means about the explanations not being based on data. We clearly agree with the reviewer statement that the interaction of vegetation, climate, and soil is context-dependent. Indeed, the purpose of this study and the analytical approach we used was to explore the context-dependence of the relationships between vegetation and soil organic matter. In doing so, we found that overall this interaction is particularly strong at warm and arid grassland sites, which is an important and novel finding.

Our main finding is now further supported by a new six-panel figure (**Fig. S2 in the Supplement**) showing that the relationships between Shannon index and SOC as well as Shannon index and C:N ratio strongly depend on climate, and are strongest in warm and arid climates.

Minor issues

- Line 69: grasslands or grassland sites We added the word sites.
- Line 82: Mislabeled references We added two more references to support the statement.
- Line 102: I recommend supplementing these grassland sites with grassland types. Our dataset comprises 19 different grassland types. We now added this information to the main text and in the

Material and Methods, and we added a column to **Table S1 in the Supplement** that displays the grassland type for each grassland site.

- Line 106: I recommend supplementing these grassland sites with grassland types *see above*

- Line 111: The relationships among plant species richness, productivity, SOC and C:N ratio across the 84 grasslands should be added at least in the supplement material. We added the requested figures showing plant species richness to the **Supplement (Fig. S6)**. In the manuscript, we show the relationship of Shannon index and SOC (**Fig. 2**), Shannon index and C:N ratio (**Fig. 3**) and plant productivity and TOC (**Fig. S3 in the Supplement**). Shannon index and plant species richness are by definition strongly correlated, so these new figures look much like those that were already in the manuscript. However, since the reviewer asked about it, we added the requested figure showing species richness to the supplement.

- Line 112: Wang et al. (2019) indicated that effect of species richness on productivity shifts from strongly positive in low-productivity communities to strongly negative in high-productivity communities. We found that the article analyzed the relationship between diversity and SOC and C:N ratio in different climatic conditions, but did not analyze the relationship between diversity and biomass in different climatic conditions to better support the conclusions.

We thank the reviewer for pointing this out. It is true that the reference to the article by Wang et al. did not fit here. The authors divided their dataset into three equally-sized groups of grassland sites according to productivity. We removed the reference from this part of the manuscript.

- Line 117-119: I am lost here. How the factor was calculated. Is it comparable at different diversity gradients

These factors from previous studies are calculated based on the organic carbon contents of the soils in the plots with only one plant species and the ones with the highest plant diversity. They are comparable since they all depict the maximum increase in SOC observed due to an increase in species number. We now reworded the sentence to make this clearer.

“In three biodiversity experiments, the maximum increase in SOC caused by the largest increase in plant diversity was by a factor of 1.2 (1, 2) and 1.7 (3).” (lines 130-132)

- Line 131: Examples of forest ecosystems are not relevant.

Fair enough. We removed this sentence and the reference from the manuscript.

- Line 116-197: In Fig.2 and Fig.3, I found that the regression analysis was fitted based on site data, not plot data, so that the role of the plot data is only to show? The smaller number of sample plots (N<17) may not support the findings of the paper. At least the error bar of each site should be shown. In addition, only a subset of the results meeting the climatic threshold conditions were shown, and another part should be shown to prevent bringing down the reader's understanding of how climate affects the diversity-SOC relationship.

We added plot-level data to the figures (**Figures 2-4 and Figures S3-S6 in the Supplement**) for transparency because it allows the reader to gain insights into the variability. From the information for authors and the policy checklist, we understand that it is the journal's preference to show individual data points in the figures rather than aggregated data. Adding error bars (for x and y) rather decreases readability of the figures and does not add additional information. Showing the actual data is our preference for transparency, however, we could add error bars, if the editor prefers.

There seems to be a misunderstanding here; the figures show all results meeting the climatic threshold conditions. We now improved the description of the data analysis (See Material and Methods **lines 329-339**) to avoid misunderstanding.

- Line 188: In Figure 4, the model is fitted with site data and the presentation of plots makes no sense. And all the plots of the same site seem to share one set of climatic data, please give the resolution of the climatic data.

This is correct, mean annual temperature and precipitation are identical for all plots of each site. We added plot-level data to the figures for transparency because it allows the reader to gain insights into variability (of the dependent variable) at each site.

- Line 216: Missing segment comma. We now added a comma.

- Line 238: “Evidence indicating that plant diversity impacts SOC through its effect on C:N ratio”. But the structural equation model in Figure 1 demonstrates that the pathway relationship from soil C: N to SOC was not significant.

That phrase appears to be taken out of context. The entire sentence reads, “Instead, we present evidence indicating that plant diversity impacts SOC through its effect on C:N ratio, in a manner that depends on climate (Fig. 1B).”

- Line 466: Materials and methods line feed Fixed

Reviewer #3: Thank you for the opportunity to review your paper. I appreciate the time and work that goes into preparing a manuscript.

This manuscript uses comprehensive data from the NutNet and focuses on the role of background climate conditions in modulating the relationship between plant diversity and soil carbon. The structural equation modelling approach was used to detect the interaction effects of climate conditions and plant diversity on soil C. As a main result, the authors showed that plant diversity is positively related to soil carbon content across 84 grassland sites and this positive relationship is more pronounced in warm and arid climates. Importantly, the authors found that the quality of organic matter, instead of the quantity of organic matter, is the main underlying mechanism of the positive relationship between plant diversity and soil carbon.

Interesting read! The design of the study is quite impressive, the results are novel, interesting, and important. However, there are some concerns. Below, I listed some major points that should be addressed.

We thank the reviewer for the positive evaluation of our manuscript!

General Comments:

1. The authors stated that real-world observation research has advantages over biodiversity experiments for exploring the BEF relationship due to long-term developments. However, I do not agree with the long-term vs short-term is the main difference between experiments and natural observations. The main differences are randomly assembled species richness levels and uniform environment in experiments. Please see van der Plas, F. Biodiversity and ecosystem functioning in naturally assembled communities. *Biol. Rev.* 94, 1220–1245 (2019).

Thus the positive relationships in experiments might be not real.

We agree with this statement and added the following sentences in the Introduction “Furthermore, in biodiversity experiments, abiotic factors are usually chosen to vary as little as possible in order to isolate the effect of plant diversity on ecosystem properties, which is opposite of real-world conditions where variation in abiotic conditions fosters plant diversity. In natural systems, variation in biodiversity is non-randomly distributed across space and time, whereas species combinations are randomly assembled in many biodiversity experiments (17).” (lines 92-96).

2. The authors state that biodiversity experiments might underestimate the effect of plant diversity on SOC since they found more strong effects. However, it's worth considering that stands with higher nutrient levels (as indicated by high SOC) could potentially support a greater number of species.

Therefore, it may be the soil nutrient levels influencing diversity rather than plant diversity directly impacting soils. In a recent study, it was found that the effect of diversity on soil carbon accumulation is positive but relatively weak.

Chen, X. Tree diversity increases decadal forest soil carbon and nitrogen accrual. *Nature*, 2023.

We found that the positive correlation of soil nitrogen content and Shannon index (**Fig. S4 in the Supplement**) was less strong than the one of SOC and Shannon index (**Fig. 2**), which likely suggests that plant diversity is more strongly related to soil organic matter than plant diversity is affected by nitrogen/nutrients derived from soil organic matter. Although we mentioned this in the manuscript before, we have reworded this part of the manuscript to make it clearer. This is the new sentence: *“We found that the positive correlation of soil nitrogen content and Shannon index (Fig. S5) was less strong than the one of SOC and Shannon index (Fig. 2), which is in accordance with earlier work (1), and suggests that plant diversity is more strongly related to soil organic matter than plant diversity is affected by nitrogen/nutrients derived from soil organic matter.”* (**lines 248-251**).

3. I very much like the new mechanism (litter C:N) that arises in this manuscript. However, it is important to note that this does not necessarily imply that only the quality, and not the quantity, of litter inputs influences SOC. It is plausible that the increased plant diversity resulted in enhanced root productivity, ultimately leading to an increase in soil carbon. Please see:

Chen, S. P. et al. Plant diversity enhances productivity and soil carbon storage. *Proc. Natl Acad. Sci. USA* 115, 4027–4032 (2018).

We agree with this comment. In fact, we found a positive relationship between plant biomass and SOC (see **line 200 and Fig. S3**), but no significant relationship between plant diversity and plant biomass across all sites. We now added the word “*aboveground*” before biomass in **lines 127, 142 and 264**.

4. I am curious about the inclusion of both site and plot data in Figure 2. Is it appropriate to pool them together? Furthermore, during the model selection process, did the authors take into account the possibility of multicollinearity?

Sorry for the confusion here. We have added the following sentence to the captions of **Figs 2-4** as well as to **Figures S3-S6 in the Supplement**. *“The linear models were plotted to the site-level data (and not to the plot data, which is shown to give insight into the variability).”*

We added plot-level data to the figures for the sake of transparency because it allows the reader to gain insights into variability within and among sites.

Specific Comments:

Line 91 please see

Chen, X., Hisano, M., Taylor, A. R. & Chen, H. Y. H. The effects of functional diversity and identity (acquisitive versus conservative strategies) on soil carbon stocks are dependent on environmental contexts. *For. Ecol. Manag.* 503, 119820 (2022). Interesting study (we did not add this paper as a reference because reviewer#1 criticized the reference to studies about forest ecosystems).

Line 151-152 Please note high plant diversity could SOC via soil microorganisms due to increased litter inputs, which is based soil microbial carbon pump hypothesis. Only increased litter C:N without any enhanced litter inputs might decrease soil activity and SOC from soil microbial residues. Please see:

Lange, M. et al. Plant diversity increases soil microbial activity and soil carbon storage. *Nat. Commun.* 6, 6707 (2015).

We agree with this statement. In **line 239** we write *“The model predicts that both the C:N ratio and plant biomass affect SOC content”*. We could only speculate how differences in C:N ratio without differences in plant biomass affect SOC content, but we have no data on this.

Reviewer #3 (Remarks to the Author):

I have reviewed this manuscript before (Referee 3), so I will take it from there.

Great to see this new version and the detailed responses to my previous comments. Thanks for that. Generally, most of my previous concerns have been satisfactorily addressed and the manuscript has clearly improved. But, I still have some concerns.

Line 104-106 The assertion that plants at arid sites produce recalcitrant litters does not necessarily imply that diverse plant communities would have high soil organic carbon content. Your speculation appears to be based on two assumptions: first, that diverse plant communities produce more litters, and second, that litters in arid areas are difficult to decompose. It would be beneficial to provide further clarification on these assumptions. Furthermore, despite your findings not aligning with the suggestion made in reference 20, where diversity effects were expected to be stronger in arid areas compared to humid area, it is essential not to dismiss this possibility in the Introduction.

Secondly, considering that plant diversity has the potential to alter the stem-to-leaf ratio, allowing for better light utilization, it is reasonable to expect that an increase in biomass would accompany higher plant diversity. This can be attributed to the complementary resource use among diverse plant species. Thus, I am not convinced by the speculation provided to explain the lack of diversity effect on biomass.

Third, Although the author suggests that the diversity of organic compounds might lead to a reduction in the decomposition rate, citing some references, it is important to note that a substantial body of research supports the contrary view. Many studies demonstrate that the chemical diversity of litter could actually increase the decomposition rate. In fact, the number of papers indicating a positive effect on decomposition rates is likely to surpass those suggesting a negative impact. So I am not convinced by this explanation, it is too speculative.

Fourth, it is crucial to consider the historical context of the study sites within the Nutrient Network. Before drawing conclusions, it is essential to ascertain whether these sites have previously undergone nutrient application. If nutrient application has occurred in the past, it could potentially introduce significant bias into the results.

Fifth, for previous comment 2, could you please explain why the stronger positive correlation of soil organic carbon content and Shannon index compared with those between soil nitrogen content and Shannon index could indicate plant diversity is more strongly related to soil organic matter than plant diversity is affected by nitrogen/nutrients derived from soil organic matter. However, it is crucial to recognize that soil organic matter encompasses more than just nitrogen; there could be a multitude of other nutrients originating from SOC that contribute to the overall soil fertility. It is essential to acknowledge the possibility that plant diversity can not increase SOC content. Instead, both variables could be influenced by common factors, such as nutrient availability and water availability. So I would suggest to tone down the conclusion.

Sixth, thanks for the response and revision based on the previous comment 4. However, I would like to suggest adding a line based on plot data. It is very confusing for the current version.

Point-by-point response to the reviewer's comments

Reviewer #3 (Remarks to the Author):

I have reviewed this manuscript before (Referee 3), so I will take it from there. Great to see this new version and the detailed responses to my previous comments. Thanks for that. Generally, most of my previous concerns have been satisfactorily addressed and the manuscript has clearly improved. But, I still have some concerns.

We thank the reviewer for reviewing our manuscript again and for the thoughtful comments, which helped us to improve the manuscript.

Line 104-106 The assertion that plants at arid sites produce recalcitrant litters does not necessarily imply that diverse plant communities would have high soil organic carbon content. Your speculation appears to be based on two assumptions: first, that diverse plant communities produce more litters, and second, that litters in arid areas are difficult to decompose. It would be beneficial to provide further clarification on these assumptions. Furthermore, despite your findings not aligning with the suggestion made in reference 20, where diversity effects were expected to be stronger in arid areas compared to humid area, it is essential not to dismiss this possibility in the Introduction.

We removed this sentence (previously lines 104-106) from the Introduction because we agree that this aspect was slightly speculative. In addition, we now reflect on this aspect in a more balanced way in the Discussion, pointing out that contradictory findings on the effect of litter diversity on decomposition have been reported (see our answer to the reviewer's third point).

Secondly, considering that plant diversity has the potential to alter the stem-to-leaf ratio, allowing for better light utilization, it is reasonable to expect that an increase in biomass would accompany higher plant diversity. This can be attributed to the complementary resource use among diverse plant species. Thus, I am not convinced by the speculation provided to explain the lack of diversity effect on biomass. It is true that plant diversity has the potential to alter the stem-to-leaf ratio. While this leads to better light availability for some plant species, it leads to increased shading for others, which can cause species loss. Thus, there are two antagonistic processes, which seem to cancel each other out. While plant diversity can increase plant biomass due to complementary use of resources by different plant species, elevated aboveground biomass can cause species loss due to shading. This explanation is based on a previous study on a subset of sites studied here showing that these two contradicting forces are acting together. We reworded this section to make it clearer, and toned it a little down. (lines 121-126).

Third, although the author suggests that the diversity of organic compounds might lead to a reduction in the decomposition rate, citing some references, it is important to note that a substantial body of research supports the contrary view. Many studies demonstrate that the chemical diversity of litter could actually increase the decomposition rate. In fact, the number of papers indicating a positive effect on decomposition rates is likely to surpass those suggesting a negative impact. So I am not convinced by this explanation, it is too speculative.

We thank the reviewer for pointing this out. We added the following sentences acknowledging contrasting findings regarding chemical diversity and decomposition rate, along with two additional references to the Discussion.

“Previous research on the relationship between chemical diversity of plant litter and decomposition reported contrasting findings (40), which might be because of the climate-dependence of the effect of litter diversity on decomposition. This is corroborated by a recent litter decomposition experiment conducted along a precipitation gradient in Chile, which found that mixing litters of several plant species led to a negative effect on decomposition only at the arid end of the precipitation gradient (41). This result supports the interpretation that in warm and arid regions, chemical diversity of litter which increases with plant species diversity causes the strong relationship between plant diversity and SOC found here.” (lines 206-2013).

Fourth, it is crucial to consider the historical context of the study sites within the Nutrient Network. Before drawing conclusions, it is essential to ascertain whether these sites have previously undergone nutrient application. If nutrient application has occurred in the past, it could potentially introduce significant bias into the results.

We now added the following information in the Material and Methods sections.

“Nine of the 84 sites, located on different continents and in different climate zones, are classified as the grassland type old field (Table S1), and they likely received some fertilizer in the past. Yet, the large majority of the sites has never received any fertilizer. Furthermore, nutrient contents differ widely among the 84 grasslands, independently of former land-use.” (lines 307-311)

For this study, we choose data that were collected in the year before any experimental treatment started, which means that the sites were not experimentally manipulated at the time of data collection. We had pointed this out in the beginning of the Material and Methods section, and we now also added this information at the end of the Introduction (lines 113-115).

Fifth, for previous comment 2, could you please explain why the stronger positive correlation of soil organic carbon content and Shannon index compared with those between soil nitrogen content and Shannon index could indicate plant diversity is more strongly related to soil organic matter than plant diversity is affected by nitrogen/nutrients derived from soil organic matter. However, it is crucial to recognize that soil organic matter encompasses more than just nitrogen; there could be a multitude of other nutrients originating from SOC that contribute to the overall soil fertility. It is essential to acknowledge the possibility that plant diversity can not increase SOC content. Instead, both variables could be influenced by common factors, such as nutrient availability and water availability. So I would suggest to tone down the conclusion.

There seems to be a misunderstanding here. The correlation with the Shannon index is higher for soil C than for soil N (not the other way around). The reviewer is, of course, right in saying that plant diversity and soil C content could be influenced by the same factors, such as nutrient availability, and nutrient availability is to some extent related to soil C. We cannot fully disentangle these feedbacks (bidirectional interactions). However, our results seem to indicate that the (indirect) effect of plant diversity on soil C is larger than the effect of soil N on plant diversity (as indicated by the stronger correlation). The indirect effect of plant diversity on soil C is likely caused by the change in quality (increase in C:N ratio) of the plant biomass with increasing plant diversity.

We now revised this part of the Discussion and we added the results of a new regression analysis, involving data on plant available phosphorus. Phosphorus can also be bound covalently in the organic matter (similarly as nitrogen and opposed to other nutrients such as potassium or magnesium) and is therefore particularly important in this context. This new part of the Discussion now reads as follows.

“In addition, we observed that phosphorus, which can also be bound in organic matter, was not significantly correlated with the Shannon index ($P = 0.914$). Together, these findings suggest that the effect of soil nutrients on plant diversity is smaller than the effect of plant diversity on SOC that is likely caused by a change in the quality of the organic matter, leading to reduced organic matter decomposition (see above).” (lines 272-276).

Sixth, thanks for the response and revision based on the previous comment 4. However, I would like to suggest adding a line based on plot data. It is very confusing for the current version.

All data analyses were conducted based on the site-level data (and not based on the plot data). Plot data is shown in Figs 2-4 in order to give insight into the within-site variability. This is pointed out in the captions of all figures. We now added an explanation for why the data analyses was conducted based on the site-level data in the Material and Methods section (*“since the observations for different plots of one site are not independent of each other”*) (lines 352-353).

In addition, we now added the information that number of observation refers to number of grassland sites in the caption of Table 1.